# Influence of Parental Attitude Toward Psychiatric Help on Their Children’s Suicidal Ideation: A Convenience Sample Study on One South Korean Middle School

**DOI:** 10.3390/ijerph17207656

**Published:** 2020-10-20

**Authors:** Yoo Mi Jeong, Hanjong Park

**Affiliations:** 1College of Nursing, Dankook University, Cheonan-si 31116, Korea; yjeong20@dankook.ac.kr; 2College of Nursing, The Catholic University of Korea, Seoul 06591, Korea

**Keywords:** adolescent, help-seeking behavior, depression, parenting, stigma, suicidal ideation

## Abstract

Depression, depression stigma, and attitude toward psychiatric help are associated factors of suicide in adolescents. As parents are the main decision-makers of receiving professional help for their children’s depression and suicide, parental factors influencing their children’s suicide should be examined. Moreover, parents’ help-seeking attitude for their own mental health problems could affect their children’s mental health problems. Therefore, this study examined the serial mediation of adolescents’ depression, depression stigma, and attitude toward psychiatric help in the relationship between parental attitude toward psychiatric help and the suicidal ideation of their children, using data of 103 parent–child pairs. A cross-sectional study was conducted by employing a self-administered survey. A serial mediation analysis was performed using Amos 25.0. Parental attitude toward psychiatric help directly and indirectly influenced children’s suicidal ideation. Children’s depression stigma, attitude toward psychiatric help, and depression mediated the relationship of parental attitude toward psychiatric help and their children’s suicidal ideation. When parents have a more positive attitude toward psychiatric help, their children’s suicidal ideation become more decreased. Enhancing only parental attitude toward psychiatric help may make a positive change on their children’s suicidal ideation. The study findings imply that when developing and applying youth suicide prevention programs, how parents affect their children’s suicidal ideation should be considered as well as adolescents’ depression stigma, attitude toward psychiatric help, and depression. Given the results of this study, healthcare providers may better evaluate the effectiveness of their intervention programs for preventing adolescents’ suicide.

## 1. Introduction

Adolescent suicide has been a serious problem for decades globally [1]. The highest suicide rate was shown globally among the 35 wealthy Organizations for Economic Cooperation and Development (OECD) countries in 2015: 25.8 per 100,000 [2]. Moreover, since 2014, the first leading cause of mortality among adolescents in South Korea has been suicide. In 2016, the major causes of deaths in adolescents were suicide (7.8%), transport accidents (3.8%), and cancer (3.1%) [3]. In addition, adolescent suicide is a serious threat to family, friends, neighbors, and community members, as well as their own personal life [4]. Therefore, various interventions and research are necessary to prevent adolescents’ suicide. Suicidal ideation can be a prior step and a warning sign for suicide attempts since most people who attempted suicide had previously fantasized about death. According to Reinherz et al. [5], the group with suicidal ideation was 12 times more likely to die by suicide than their counterparts with non-suicidal ideation. Since suicidal ideation is a predictor of suicide attempts [6,7], suicidal ideation in adolescents should be explored for early detection of a suicide attempt. In addition, suicidal ideation in adolescents can cause the prevalence of lifetime suicidal ideation and attempt [8]. Therefore, it is important to identify factors related to suicidal ideation and pay attention to suicidal ideation as a preventive approach to adolescent suicide. 

Untreated depression is one of the major causes of suicidal behavior; 56% of suicidal deaths were caused by untreated psychiatric disorders, including depression, in South Korea [3,9]. According to the World Health Organization (WHO), depression has been recognized as a major mental health problem [10] and it usually begins during adolescence [11,12]. Depression in adolescence is associated with high stress; Korean adolescents aged between 11 and 14 reported significant conflict-related stress from their parents [13]. The parents’ pressure on their children causes school failure, delinquency, and vocational problems due to depressive symptoms [13]. Korean high school students are under high academic stress to enter good universities, especially the top three prestigious universities or any medical school in Korea [13]. Entering a good university is often tied to family prestige and honor so that many Korean adolescents take their own lives out of the family issues and educational pressure [13,14,15]. This is not a problem that only happens in Korea. According to a study conducted in the United Kingdom, the academic pressure placed on children in the family is more likely to produce severe depressive symptoms among adolescent children [16].

Although depression has been acknowledged as a clinical condition, it is still often perceived as a shameful, unproductive, and socially undesirable mental illness [17], and this perceived stigma is a major barrier for patients seeking mental healthcare for depression [18,19,20,21,22]. Adolescents’ depression stigma was reported as a predictor of attitude toward psychiatric help [23]. The negative or absence of a positive attitude toward psychiatric help in adults is an important factor related to suicide behavior [24]. Attitude toward psychiatric help can be defined as attitudes toward behaviors that seek help by seeking professional institutions, such as counseling, treatment, and psychiatric treatment in the event of mental health issues [25]. According to a survey regarding mental disorders in Korea [26], mental health services usage by people with any kind of disorder was 16.6%, and 40.4% of them with mood disorders visited psychiatrists, which is relatively low compared to other OECD countries [27,28] Attitude toward psychiatric help for depression has been rigorously studied for exploring facilitated or barrier factors in many countries, to help adolescents in suicidal ideation use professional help at the right time.

It would be important to understand the attitude toward psychiatric help of adolescents thinking about and attempting suicide, and how the attitude toward psychiatric help act as a factor influencing suicidal ideation and attempts. In terms of a parental help-seeking attitude for their children’s mental health problems, previous studies reported that parental attitude toward psychiatric help for their children was influenced by parents’ recognition of their adolescent children’s mental health problems [29], their past experience of seeking help for their children’s mental health problems [4], and health belief about their children’s mental health problems [4]. As parents are making decisions for their children’s use of mental health services, it would be critical to understand parental attitude toward psychiatric help for their own mental health issues because this attitude would be similarly applied when their children have such an issue (i.e., depression). In addition, consideration of both adolescents and their parents’ factors is necessary to develop educational programs that may effectively prevent suicidal ideation or suicide attempts of children. However, there is limited information on the influence of parental attitude toward psychiatric help for their children’s suicidal ideation, the dynamic of depression stigma, attitude toward psychiatric help, depression, and suicidal ideation among adolescents. Moreover, the previous studies were focused on parental attitude toward psychiatric help for their children’s mental health problems, not on their help-seeking attitude towards their own mental health problems. Therefore, research on how parents’ attitudes toward the mental health services for their own mental health issues influence their adolescent children’s suicidal ideation needs to be conducted.

The purpose of this study was to examine the factors affecting suicidal ideation of children based on a serial mediation model using a parent–child pair data set. Based on the previous studies, we hypothesized that parental attitude toward psychiatric help for their own mental health issue would influence the relationship among depression stigma, attitude toward psychiatric help, depression, and suicidal ideation of children. The specific aims of the study were as follow: (a) to examine the correlations among the variables of the parents and children, including depression stigma, attitude toward psychiatric help, depression, the suicidal ideation of children, and parental attitude toward psychiatric help; (b) to examine the serial mediation of adolescents’ depression stigma, attitude toward psychiatric help, and depression in children in the relationship between parental attitude toward psychiatric help and their children’s suicidal ideation.

## 2. Materials and Methods

### 2.1. Sampling and Data Collection

This was a cross-sectional study. Participants of this study were recruited from a middle school located in the southern part of South Korea. Since the number of samples needed for statistical significance is five to ten times that of the parameters in the path analysis, at least 100 participants were needed to be acceptable [30]. The response rate was 95% since 103 out of the 120 distributed survey questionnaires were collected. Data were collected from 2018 June to August using a self-administered survey. After receiving approval of the Institutional Review Board (IRB), a teacher in the middle school and the primary investigator (PI) gave an invitation and recruitment letter to all students in the middle school. The consent form and questionnaires were enveloped with a stamp and sent to children and their parents who agreed to participate in this study. The consent form included a statement about voluntary participation and protection of anonymity and confidentiality. The PI obtained written consent from all the participants, including both parents and their children.

### 2.2. Instruments

For both parents and their children, the questionnaire of attitude toward psychiatric help was distributed. For adolescents, the questionnaire of suicidal ideation, depression, and depression stigma was distributed. 

#### 2.2.1. Attitude toward Psychiatric Help of Parents and Children

To measure attitude toward psychiatric help, the 10-item “Attitudes toward seeking psychiatric help” (ASPH) survey [31] with a five-point Likert scale was used for measuring attitudes toward seeking professional help for depression [32]. A higher total sum of scores indicated more positive attitudes toward seeking professional help. Items 2, 4, 8, 9, and 10 were reverse-scored and the scores of the 10 items were summed [32]. During development, the Cronbach’s alpha in adults was 0.76 [32] and the current study’s alphas for the ASPH of parents and their children were 0.72 and 0.76.

#### 2.2.2. Suicidal Ideation of Children 

Suicidal ideation of children was measured by the Suicidal Ideation Questionnaire (SIQ) with 30 items [33]. Examples of the items included (1) “I thought about suicide”; (2) “I thought about when to die by suicide”; and (3) “I thought I had no reason to live anymore”. Items were rated on a seven-point Likert scale and the total sum of these items was used [33]. A higher score indicates a higher level of suicidal ideation. The original SIQ was purchased from the Psychological Assessment Resource, Inc. (Florida, FL, USA) [34] and the use of the Korean-version of SIQ (2018) was permitted by the Korean Educational Development Institute [35]. The Cronbach’s alpha for the SIQ in children was 0.99. 

#### 2.2.3. Depression in Children

The Beck Depression Inventory-II (BDI-II) was used for detecting depression, which consisted of 21 out of the 22 items [36]. The suicidal ideation item of the BDI-II was excluded to avoid multicollinearity between the BDI-II and SIQ scores. Each item was selected on a 0 to 3-point scale and the scores of each item were summed and scored. The range of the total score is 0 to 63 points: 14 and over points indicate depression [37]. BDI-II addresses depressive symptoms, including weight loss, change in body image, somatic symptoms, loss of energy, sleep loss, appetite loss, as well as sadness and loss of interests [36]. The Korean version of BDI-II (K-BDI-II) was purchased from the Korean Psychology Association [38]. The Cronbach’s alpha of the K-BDI-II in adolescents was 0.95.

#### 2.2.4. Depression Stigma of Children

To measure the stigma associated with depression (i.e., personal stigma (6 items) and public stigma (9 items), the revised Depression Stigma scale was used for this study [39,40]. The 15 items used for this study were adapted from the Depression Stigma scale developed by Griffiths and colleagues [39] and were permitted to be used by the developer via email. Examples of the personal depression stigma items included: (1) “I think people with depression could snap out of it if they wanted”; (2) “I think depression is a sign of personal weakness”; and (3) “I think depression is not a real medical illness”. Examples of the public depression stigma items were (1) “Most people would not tell anyone if they had depression”; (2) “Most people would not employ someone they knew had been depressed”; and (3) “Most people would not vote for a politician they knew had been depressed”. Each item was measured on a scale ranging from 0 to 4, and the total score from 0 to 60; higher scores indicated greater stigma. The Cronbach’s alpha for the adapted Depression Stigma scale among children was 0.86.

### 2.3. Ethical Considerations

Before recruitment began, the study was approved by the University IRB (Approval No. KYU-2018-059-01). The PI obtained written consent from all the participants, including both parents and their children. Parents were able to choose the way to receive their questionnaire with a returned envelop between getting the questionnaire by mail and getting the questionnaire through their children. Answered questionnaires by the parents were directly returned to us. Therefore, children were not asked to bring their parents’ answered questionnaire. 

### 2.4. Data Analysis

Data were analyzed using SPSS version 25.0 and Amos 25.0 (IBM Corp., Armonk, NY, USA). The Pearson Correlation Coefficient was conducted to examine the associations among the variables of parents and children. A serial mediation analysis of the parent–child pair data was performed using the Maximum Likelihood (ML) procedure. The decision on the model fit was based on *χ*^2^ statistics, the Comparative Fit Index (CFI), Tucker–Lewis Index (TLI), and Root Mean Square Error of Approximation (RMSEA). Good model fit is defined as insignificant *χ*^2^ statistics, a CFI > 0.95, TLI > 0.95, and RMSEA < 0.06 [41]. The purpose of using serial mediation analysis in this study was to better understand the dynamics among the variables by including an examination of indirect effects of the exogenous variables on the endogenous variable. To examine the indirect effect of parental attitude toward psychiatric help on the suicidal ideation of children with serial mediators, including depression stigma, attitude toward psychiatric help, and depression in children, bootstrapping was conducted with 95% confidence intervals from 10,000 bootstrap samples. If a zero is not included in the 95% confidence interval, it presents a significant indirect effect at the significance of alpha at 0.05. 

## 3. Results

### 3.1. Sample Characteristics

For this study, 103 coupled data sets, paired as the Korean parents and their children, were collected. The majority of children participants were girls (93.2%) and second (49.5%) or third grade (32.0%) middle school students. Approximately 83% of them were first- or second-born children. The mean age of the parents was 44 years old. The majority of parents were mothers (92.2%) and possessed a college degree or higher (72.8%) (Table 1). According to the national statistics regarding college education in South Korea, the enrollment rate in tertiary education was about 52.5% in 2000 and about 67.6% in 2019 [42]. These statistics would indicate that highly educated parents were not selectively included in this study.

### 3.2. Correlations among Variables of Parents and Children

Table 2 presents the correlations among the variables of the parents and their children. Suicidal ideation of children was significantly associated with depression in children (r = 0.81, *p* < 0.001), children’s attitude toward psychiatric help (r = −0.36, *p* < 0.001), and parental attitude toward psychiatric help (r = −0.21, *p* = 0.032). Children’s depression was also significantly associated with their attitude toward psychiatric help (r = −0.43, *p* < 0.001). Additionally, children’s attitude toward psychiatric help was significantly associated with their depression stigma (r = −0.26, *p* = 0.001). Depression stigma of children was negatively associated with parental attitude toward psychiatric help (r = −0.22, *p* = 0.027).

### 3.3. Mediating Effects of Depression Stigma, Attitude toward Psychiatric Help, and Depression in Children in the Relationship between Parental Attitude toward Psychiatric Help and the Suicidal Ideation of Their Children 

The mediating effects of depression stigma, attitude toward psychiatric help, and depression in children in relation to parental attitude toward psychiatric help and the suicidal ideation of their children are reported in Table 3 and Figure 1. The serial mediation model fit the data well using the ML estimation: *χ*^2^ (df = 5) = 7.126, *p* = 0.211; CFI = 0.985; TLI = 0.970; and RMSEA = 0.065. As indicated, the significance of the indirect effect was presented based on the bootstrapping approach (Table 3). Parental attitude toward psychiatric help has alleviating effects on suicidal ideation of children (β = −0.115, *p* = 0.046) and depression stigma of children (β = −0.218, *p* = 0.024). As children’s stigma was high, their attitude toward psychiatric help was low (β = −0.256, *p* = 0.008). Children’s attitude toward psychiatric help also has an alleviating influence on the depression in children (β = −0.427, *p* < 0.001). In addition, as depression in children was high, their suicidal ideation was also high (β = 0.804, *p* < 0.001).

The direct and indirect effects of parental attitude toward psychiatric help were examined; it was observed that the direct effect of parental attitude toward psychiatric help is significant (direct effect = −1.055; bootstrapped lower lever confidence interval [bootLLCI], bootstrapped upper level confidence interval [ULCI]: −2.226, −0.117). As a result of verifying the serial mediating effect, Parental attitude toward psychiatric help (x) → Depression stigma of children (m1) → Children’s attitude toward psychiatric help (m2) → Depression of children (m3) → Suicidal ideation of children (y) is −0.568 (bootLLCI, bootULCI: −0.568, −0.024), which is considered to be statistically significant (Figure 1 and Table 3). Thus, the serial mediation effect of depression stigma, attitude toward psychiatric help, and depression in children are confirmed in the relationship between parental attitude toward psychiatric help and the suicidal ideation of children.

## 4. Discussion 

In this study, we could identify the direct and indirect influence of parental attitude toward psychiatric help on their children’s suicidal ideation. We also explored the significant mediating effect of depression stigma, attitude toward psychiatric help, and depression in children in the relationship between parental attitude toward psychiatric help and their children’s suicidal ideation.

First, parental attitude toward psychiatric help played a positive role in directly lowering the children’s suicidal ideation. When parents had a higher positive attitude toward psychiatric help for mental healthcare services, their children showed lower suicidal ideation. This finding was partially consistent with a previous study showing the significant effects of parenting style on increasing children’s suicidal ideation [43]. The present study suggests that parental attitude toward psychiatric help positively contributes to the mechanism of alleviating children’s suicidal ideation. Therefore, including assessments of and enhancing attitudes toward psychiatric help of not only the children but also their parents for mental health services may be pivotal and cost-effective to make positive changes on the children’s suicidal ideation.

Secondly, the results of this study confirmed that the significant association with suicidal ideation was strong in the order of depression, children’s attitude toward psychiatric help, and parental attitude toward psychiatric help. Although the depression stigma of children was not directly related to suicidal ideation, depression stigma of children indirectly influenced suicidal ideation in the serial mediation model of this study. Children’s suicidal ideation was positively influenced by their depression and this depression showed a mediating effect between children’s attitude toward psychiatric help and their suicidal ideation. This result was consistent with Jeong and Yi’s study [44], who reported the mediating effect of depression on the relationship of attitude toward psychiatric help and suicidal ideation among 151 adolescents in South Korea. These results also are in line with prior studies that have shown that people with depressive disorders are at least 40 times more likely to die by suicide, and the negative perceptions of psychiatric treatment were negatively related to the treatment of depression [45,46]. Adolescents with suicidal ideation are more likely to be depressed [42] and depressed adolescents seek professional help less than those without depression [47]; it is like they get trapped in a vicious circle. Therefore, this study suggests that accurate assessment and intervention of depression be a priority rather than only making adolescents’ attitudes toward psychiatric help more positive, for overall decrease and prevention of adolescent suicide. As previous studies [45,48] suggested, concrete relationships among schools, hospitals, and community systems should be established and parents should be educated about the importance of using professional help for their own and their children’s mental health problems. 

Thirdly, one of the strengths of this study was that we included parents and their children’s attitudes toward psychiatric help simultaneously in our mediation model. This study showed that it is necessary to pay attention to parental attitude toward psychiatric help to enhance the reduction of their children’s depression stigma and to improve the children’s attitude toward psychiatric help. When parents showed a negative attitude toward psychiatric help, it influenced their children’s depression stigma and then this stigma negatively influenced the children’s attitudes toward psychiatric help. The findings of this study are also consistent with a study that reported depression stigma was associated with depression, along with the use of psychiatric help, and that depression stigma could impede individuals from seeking treatment [22]. The results of this study are also aligned with Pargas et al.’s study [43] in that parental interest in their children’s depressive symptoms would affect the early and proper decision-making of seeking professional help for their children’s depression [43]. In addition, adolescents’ attitude toward psychiatric help was significantly influenced by the stigma of their peers and teachers [39,47,49,50,51]. According to Lee et al. [52], adults’ depression was shown to mediate the relationship between depression stigma and attitude toward psychiatric help. Thus, we suggest assessing not only how adolescents perceive the attitudes of their parents, peers, and teachers toward depression and their treatments, but also the parental depression stigma needs to be examined when exploring adolescents’ attitudes toward psychiatric help. 

Regarding the examination of adolescents’ suicidal ideation, previous studies used either adolescents’ or their parents’ reports about depression stigma, depression, and attitude toward psychiatric help. The findings of this study may provide practical implications for adolescents and their parents. First, parental education is necessary to make their own attitude toward psychiatric help positive. This change would decrease their children’s depression stigma and hence the suicidal ideation of their children. Secondly, it would be necessary for adolescents to consider how depression stigma, depression, and attitude toward psychiatric help may affect their suicidal ideation. 

However, the generalizability of this study was limited since the data were collected in one middle school located in a southern part of South Korea, using convenience sampling. Secondly, most participants (both parents and their children) were female so that the findings of this study might be influenced by sampling bias. Therefore, for future studies we need to recruit more fathers and sons. Fourthly, all instruments were self-reported so that social desirability or recall bias may limit the study findings. Thirdly, the study design was a cross-sectional, precluding explanation of causal relationships among variables. Fourthly, this study measured the parental attitude toward psychiatric help for their own help, so that it might be limited in exploring the parental attitude toward their children’s mental health issues. Lastly, family dynamics and function as well as parental depression stigma should be considered to examine adolescents’ attitudes toward psychiatric help. 

### 4.1. Recommendation for Future Research

For future research, a larger sample size with longitudinal data targeting middle and high school students from various regions should be conducted to increase the generalizability of the study findings. Secondly, diverse family factors that may be related to adolescents’ depression and suicidal ideation—such as parental communication skills, parenting styles, stress, stress coping styles, and family function—need to be explored to better examine the influence of parents on their children’s suicidal ideation. Thirdly, since gender differences could contribute to measures of depression, attitude toward psychiatric help and suicidal ideation [53] in relation to the gender differences of the parents and their children regarding children’s suicidal ideation should be explored. Fourthly, all instruments were self-reported so that social desirability or recall bias may limit the study findings. 

### 4.2. Implications for Clinical Practice

When parents have a positive attitude toward psychiatric help, their children’s suicidal ideation may decrease. Therefore, educational programs for parents having adolescent children should be established to enhance their attitude toward psychiatric help. Enhancing parental attitude toward psychiatric help may make positive changes in their children’s suicidal ideation. Moreover, for adolescents, we suggest the following recommendations. First, early detection of depression, the closest antecedent of suicidal ideation or behavior, would be pivotal through accurate and regular check-ups. Secondly, healthcare providers also need to assess existing depression stigma and attitude towards psychiatric help and hence develop educational plans to change them. Thirdly, health providers should develop an intervention program that takes into account the dynamics of parent–child relationships and needs to assess the effectiveness of the intervention program by analyzing parent–child pair data.

## 5. Conclusions

The findings of this study suggest efforts are needed to offer assessment and intervention programs to decrease children’s suicidal ideation using parent–child pair data. Since parental attitude toward psychiatric help directly and indirectly influenced their children’s suicidal ideation, youth suicide prevention programs should entail the ways to enhance parental attitude toward psychiatric help. Regular assessment of depression by teachers and parents and education to improve parents’ and adolescents’ attitudes toward psychiatric help-seeking from professional mental health providers should be considered in developing youth suicide prevention programs. 

## Figures and Tables

**Figure 1 ijerph-17-07656-f001:**
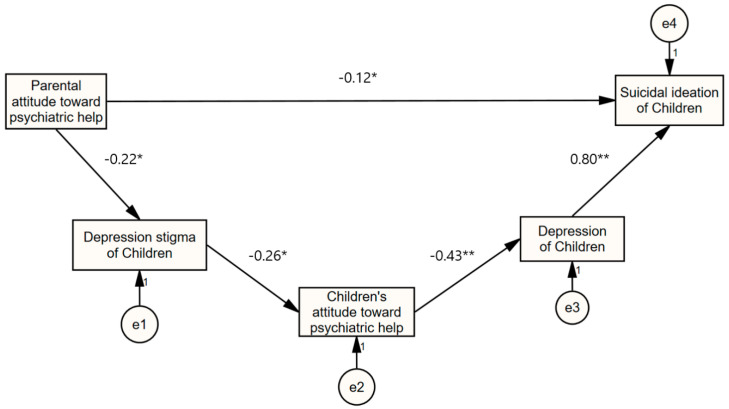
A serial mediation analysis: Mediating effects of depression stigma, attitude toward psychiatric help, and depression in children in the relationship between parental attitude toward psychiatric help and the suicidal ideation of their children (*n* = 103 parent–child pairs); * *p* < 0.05, ** *p* < 0.001.

**Table 1 ijerph-17-07656-t001:** Characteristics of participants (*n* = 103 parent–child pairs).

Characteristics		Mean (±SD) or *n* (%)
Children	Age (year)	14.11 (0.73)
	Gender	
	Boys Girls	7 (6.8) 96 (93.2)
	Grade	
	1 2 3	19 (18.4) 51 (49.5) 33 (32.0)
	Birth order	
	1 2 ≥3	44 (42.7) 41 (39.9) 13 (12.7)
	Missing	5 (4.9)
Parents	Age (years)	44.01 (3.88)
	Gender	
	Male Female	8 (7.8) 95 (92.2)
	Education	
	Middle school High school College or graduate school Missing	3 (2.9) 24 (23.3) 75 (72.8) 1 (1.0)

Note: SD = standard deviation.

**Table 2 ijerph-17-07656-t002:** Correlations among variables (*n* = 103 parent-child pairs)

Variable		Mean (SD)	1	2	3	4
1	Suicidal ideation of the children	30.65 (42.34)				
2	Depression in children	11.27 (11.11)	0.81 **			
3	Children’ attitude toward psychiatric help	34.80 (5.42)	−0.36 **	−0.43 **		
4	Depression stigma of children	40.54 (9.40)	−0.06	0.05	−0.26 **	
5	Parental attitude toward psychiatric help	36.08 (4.57)	−0.21 *	−0.12	0.13	−0.22 *

Note: * *p* < 0.05, ** *p* < 0.001; SD = standard deviation.

**Table 3 ijerph-17-07656-t003:** Serial mediation of depression stigma, attitude toward psychiatric help, and depression toward suicidal ideation among children (*n* = 103 parent–child pairs).

Effect	B	Se	Boot	BootLLCI	BootULCI
Direct effect	−1.055	0.528	0.534	−2.226	−0.117
Indirect effect					
x → m1 → m2 → m3 → y	−0.568		0.126	−0.568	−0.024
Total effect	−1.230		0.543	−2.393	−0.255

Note: Number of bootstrap samples for bias-corrected bootstrap confidence intervals: 10,000. Level of confidence for all confidence intervals: 95%. x, Parental attitude toward psychiatric help; m1, Depression stigma of children; m2, Children’s attitude toward psychiatric help; m3, Depression in children; y, Suicidal ideation of children. BootLLCI, Bootstrapped lower level confidence interval; BootULCI, Bootstrapped upper level confidence interval.

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
