# Peer review of "Influence of Parental Attitude Toward Psychiatric Help on Their Children’s Suicidal Ideation: A Convenience Sample Study on One South Korean Middle School"

_ijerph, 2020, doi:10.3390/ijerph17207656_

Round 1

Reviewer 1 Report

Thank you for the opportunity to review this manuscript regarding the effect of parental attitude toward psychiatric help on their child's depression and suicidal ideation.

  • General:

    • the term 'commit suicide' is no longer acceptable given the connotation of crime, please change to 'die by suicide' or similar. https://www.who.int/mental_health/prevention/suicide/resource_media.pdf
    • 'suicidal attempt' should be 'suicide attempt' throughout
    • There are English language problems throughout. The manuscript would benefit from editing by a native English speaker.

    Introduction:

    • Page 1, Line 31 - add citation re global adolescent suicides
    • Page 1, Lines 33-35 - move information about OECD up to second line so that structure moves from general (global suicide) to more speciic (South Korea).
    • Page 1, line 38 ' a warning sign FOR A suicidE attempt'
    • Page 1, line 43. The sentence beginning 'In addition, suicidal ideation' does not make sense and needs rewording. There are several similar issues throughout, which I do not comment upon given their number.
    • Page 2, line 51,21 - The estimate that visits to psychiatrists by people with mood disorders is 40.4% seems very high. Is this correct? This also needs a citation i.e., regarding OECD countries.
    • I think that the Introduction needs to be restructured. It seems to me that page 2, lines 66-91 might better belong before lines 46 - 65.
    • There could be better justification for the ordering of variables in the serial mediation analysis.

    Methods:

    • A short explanation of the purpose of serial mediation analysis would be useful. What is it used for and why did you choose it?
    • I wonder why depression stigma of parents wasn’t also measured? A line explaining why this wasn’t included could be useful.

    Results:

    • I initially thought that the high rate of parents (primarily mothers) with College+ education was very high and therefore that the sample was very biased. I see on doing a quick Google search that actually the rate of College education in South Korea is extremely high. I wonder if it is worth noting an approximate national statistic regarding College education in the results (a single sentence would do), so that people like me don’t assume that this sample is biased toward highly educated women and therefore not generalisable.

    Discussion:

    • Page 7, line 233 beginning ‘With regard to the relationship…’ – the way this is worded makes it sounds as though you are reporting on the results of your study, but I assume this is a finding from the literature?
    • A discussion of the relative strength of associations would be a useful addition.
    • Page 8, lines 281-285 – it may also be worth noting as a limitation that ‘parental’ attitudes measured are primarily those of mothers and therefore the effects of paternal attitudes have not really been tested. I realise this is addressed in the ‘future research’ section, but this should be mentioned in the limitations first.

Author Response

Responses to reviewer #1

Manuscript ID : ijerph-911067

Title: Influence of parental attitude toward psychiatric help on their children’s suicidal ideation: a serial mediation model

Comments and Suggestions for Authors

Thank you for the opportunity to review this manuscript regarding the effect of parental attitude toward psychiatric help on their child's depression and suicidal ideation.

  • General:
  1. the term 'commit suicide' is no longer acceptable given the connotation of crime, please change to 'die by suicide' or similar. https://www.who.int/mental_health/prevention/suicide/resource_media.pdf
    • Thank you for your valuable comment. We changed the term as “die by suicide”.

    1. 'suicidal attempt' should be 'suicide attempt' throughout
  • We revised the term as ‘suicide attempt’ throughout the whole paper.

    1. There are English language problems throughout. The manuscript would benefit from editing by a native English speaker.
  • As you recommended, we received advice from an English language professional.

Introduction:

    1. Page 1, Line 31 - add citation re global adolescent suicides
  • We added the reference about global adolescent suicides. (Line 32)
  • <reference>
  1. World Health Organization (WHO) Preventing suicide: A global imperative. Available online: http://www.who.int/mental_health/suicide-prevention/world_report_2014/en/ (accessed on 26 September 2020).
    1. Page 1, Lines 33-35 - move information about OECD up to second line so that structure moves from general (global suicide) to more specific (South Korea).
  • We reorganized the sentences as you recommended. (Lines 33-36)
    1. Page 1, line 38 ' a warning sign FOR A suicidE attempt'
  • We revised it as you recommended. (Line 39)
    1. Page 1, line 43. The sentence beginning 'In addition, suicidal ideation' does not make sense and needs rewording. There are several similar issues throughout, which I do not comment upon given their number.
  • We revised all English grammatical errors as you recommended. we got advice from an English language professional.

Ex)

In addition, suicidal ideation in adolescents can affect throughout whole life of their own.

  • In addition, suicidal ideation in adolescents can cause the prevalence of lifetime suicidal ideation and attempt. (Lines 43-44)
    1. Page 2, line 51,21 - The estimate that visits to psychiatrists by people with mood disorders is 40.4% seems very high. Is this correct? This also needs a citation i.e., regarding OECD countries.
  • We clarified that numbers and added more references about Korea and OECD data as below.

  • <References>
  1. National Institute of Mental Health Major Depression. Available online: https://www.nimh.nih.gov/health/statistics/major-depression.shtml#part_155721 (accessed on 30 September 2020).
  2. Organisation for Economic Co-operation and Development (OECD). Focus on Health: Making Mental Health Count. Available online: https://www.oecd.org/els/health-systems/Focus-on-Health-Making-Mental-Health-Count.pdf (accessed on 26 September 2020).
    1. I think that the Introduction needs to be restructured. It seems to me that page 2, lines 66-91 might better belong before lines 46 - 65.
  • We reorganized the order of paragraphs as suggested

    1. There could be better justification for the ordering of variables in the serial mediation analysis.
  • As you recommended, we revised the explanation of ordering of variables.
  • Based on the previous studies, we hypothesized that parental attitude toward psychiatric help for their own mental health issue would influence on the relationship among depression stigma, attitude toward psychiatric help, depression and suicidal ideation of children

Methods:

    1. A short explanation of the purpose of serial mediation analysis would be useful. What is it used for and why did you choose it?
  • As advised, we added a short explanation of the purpose of serial mediation analysis (Lines 171-173).
  • The purpose of using serial mediation analysis in this study was to better understand the dynamics among variables by including examination of indirect effects of the exogenous variables on the endogenous variable

    1. I wonder why depression stigma of parents wasn’t also measured? A line explaining why this wasn’t included could be useful.
  • Thank you for your suggestion. We are planning to add parental depression stigma as one of variables for the further research.

Results:

    1. I initially thought that the high rate of parents (primarily mothers) with College+ education was very high and therefore that the sample was very biased. I see on doing a quick Google search that actually the rate of College education in South Korea is extremely high. I wonder if it is worth noting an approximate national statistic regarding College education in the results (a single sentence would do), so that people like me don’t assume that this sample is biased toward highly educated women and therefore not generalisable.
  • Yes, college education attainment rate in South Korea is high. As advised, we added national statistic regarding College education in South Korea in the results section as below. (Lines 184-187) Thank you very much for the valuable comments.
  • According to the national statistics regarding college education in South Korea, the enrollment rate in tertiary education was about 52.5% in 2000 and that was about 67.6% in 2019 [42]. These statistics would indicate that highly educated parents were not selectively included in this study.
  • < Reference >
  1. Korean Educational Development Institute (KEDI) Brief statistics on Korean Education Available online: https://kess.kedi.re.kr/eng/publ/view?survSeq=2019&publSeq=4&menuSeq=0&itemCode=02&language=en# (accessed on 26 September 2020).

Discussion:

    1. Page 7, line 233 beginning ‘With regard to the relationship…’ – the way this is worded makes it sounds as though you are reporting on the results of your study, but I assume this is a finding from the literature?
  • We clarified that the result came from the previous study. (Lines 243-244)

This finding was partially consistent with the previous study that the significant effects of parenting style on the increased children's suicidal ideation existed [43].

    1. A discussion of the relative strength of associations would be a useful addition.
  • As you recommended, we added the relative strength of associations in the discussion. (On lines 249-253)
  • Results of this study confirmed that significant association with suicidal ideation was strong in the order of depression, children’s attitude toward psychiatric help, and parental attitude toward psychiatric help. Although depression stigma of children was not directly related to suicidal ideation, depression stigma of children indirectly influenced suicidal ideation with the serial mediation model of this study.

    1. Page 8, lines 281-285 – it may also be worth noting as a limitation that ‘parental’ attitudes measured are primarily those of mothers and therefore the effects of paternal attitudes have not really been tested. I realise this is addressed in the ‘future research’ section, but this should be mentioned in the limitations first.
  • We added that limitation section as well. Thank you for your valuable comments. (Lines 293-295)
  • Second, most participants (both parents and their children) were female so that the findings of this study might be influenced by sampling bias.

Reviewer 2 Report

Language/Grammar:

There are grammatical and typographical errors throughout the manuscript and several instances where the language used does not accurately convey the authors' meaning. I have not listed every single instance here but for example, the sentence in lines 35-37 stating that "adolescent suicide is a serious threat to family, friends, neighbors" etc is baffling to me, as is line 43 which states "suicidal ideation in adolescents can affect through whole life of their own." In my opinion, this manuscript would benefit from a thorough English language editing and proofreading.

Methodology/Ethics:

The authors stated that ethical approval was obtained for this study. I am concerned about whether any follow-up was done if the surveys detected the presence of depressive symptoms or suicidal ideation in the schoolchildren -- was any generic advice given to the children or the parents on obtaining help, or were the participants traced and contacted with advice on this?

Content:

It was not clear from the abstract that the authors were talking about parents' attitudes to receiving psychiatric help for their own mental health problems -- it could be easily misunderstood from the abstract that the authors are talking about parents' attitudes towards their children receiving psychiatric help. The main point expressed in lines 61-65 is not conveyed in the abstract.

Lines 69-70 -- The use of a semicolon implies that there a connection between adolescent depression and stress and pressure from parents, but I wonder if the authors are making an assumption here about the link?

Lines 74-75 -- It is not clear how the authors came to the conclusion that "many Korean adolescents take their own lives out of family pressure" based on references 16 and 17, in which I could not find any mention of the association between family pressure and adolescent suicide.

Lines 89-90 -- The authors seem to be contradicting themselves here as now they are talking about parental attitude toward psychiatric help for their children rather than for themselves. Which one did they explore in their survey?

Under Materials and Methods (lines 100-151), it can be more clearly stated which instruments were given to the parents and which to the children.

I like the visual representation of the serial mediation analysis in Fig 1; it illustrates more clearly the ideas that the authors are trying to convey. Please note the typo ("ideatin") in the top right box.

Line 205 and line 241 -- "positively affects" and "positively influenced" are ambiguous as "positive" could have opposite meanings depending on how one reads the phrase. I would suggest reviewing the wording.

Line 251 -- likewise, the meaning of "increasing adolescents' attitude" is unclear.

Lines 252-254 -- The meaning of this sentence is unclear; I don't understand how reference 39 supports the conclusion that "parents should be educated about..."

Lines 281-285 -- I think the authors have not adequately considered or discussed the limitations of their study here: they should include points such as how more than 90% of their study respondents (both the children and the parents) were female and also consider how sampling bias could influence their findings.

Lines 292-294 -- Can the authors cite references to support their statement about gender differences?

Lines 294-296 -- This content belongs in the discussion of the study limitations in the previous paragraph.

Reference 39 seems to be unavailable in the databases I searched?

Author Response

Manuscript ID : ijerph-911067

Title: Influence of parental attitude toward psychiatric help on their children’s suicidal ideation: a serial mediation model

  1. Language/Grammar:

There are grammatical and typographical errors throughout the manuscript and several instances where the language used does not accurately convey the authors' meaning. I have not listed every single instance here but for example, the sentence in lines 35-37 stating that "adolescent suicide is a serious threat to family, friends, neighbors" etc is baffling to me, as is line 43 which states "suicidal ideation in adolescents can affect through whole life of their own." In my opinion, this manuscript would benefit from a thorough English language editing and proofreading.

  • As you recommended, we got advice from an English language professional.

  1. Methodology/Ethics:

The authors stated that ethical approval was obtained for this study. I am concerned about whether any follow-up was done if the surveys detected the presence of depressive symptoms or suicidal ideation in the schoolchildren -- was any generic advice given to the children or the parents on obtaining help, or were the participants traced and contacted with advice on this?

  • We distributed brochures explaining about depression and its treatment, and the community resources about mental healthcare services to all participants in this study. Also, the program investigator distributed the contact number and asked for contacting the program investigator directly via email or phone if they have any questions related to their or their children’s depression and its treatment. In addition, as a follow-up research, we are conducting an intervention research for adolescents who scored more than 14 based on BDI-II questionnaires.

Content:

  1. It was not clear from the abstract that the authors were talking about parents' attitudes to receiving psychiatric help for their own mental health problems -- it could be easily misunderstood from the abstract that the authors are talking about parents' attitudes towards their children receiving psychiatric help. The main point expressed in lines 61-65 is not conveyed in the abstract.
  • As you recommended, we addressed the main point about parental own attitude for their own mental health problems should be studied because it can influence on their children’s mental health problems. (Abstract on lines 14-15)
  • Moreover, parental own help-seeking attitude towards their own mental health problems could influence on their children’s mental health problems.

  1. Lines 69-70 -- The use of a semicolon implies that there a connection between adolescent depression and stress and pressure from parents, but I wonder if the authors are making an assumption here about the link?
  • We addressed this link from the previous study about “Middle school students, however, were more likely to report conflict-related factors, including conflicts with parents or teachers, as major stressors. This result accords with prior studies showing that perceived conflicts with parents peak in early and middle adolescence, and thereafter decrease.”[13] We made it more clearly by changing “pressure” to “conflict-related stress”
  • Depression in adolescence is associated with high stress; Korean adolescents aged between 11 and 14 reported significant conflict-related stress from their parents [13]. (Lines 50-51)
  • <Reference>
  1. Park, S.; Jang, H.; Lee, E.-S., Major Stressors among Korean Adolescents According to Gender, Educational Level, Residential Area, and Socioeconomic Status. International Journal of Environmental Research and Public Health 2018, 15 (10), 2080-2080.
  2. Lines 74-75 -- It is not clear how the authors came to the conclusion that "many Korean adolescents take their own lives out of family pressure" based on references 16 and 17, in which I could not find any mention of the association between family pressure and adolescent suicide.
  • We added more reference on this sentence to make the connection between “wanting to die” and “family issue and educational pressure”.

Entering a good university is often tied to family prestige and honor so that many Korean adolescents take their own lives out of the family issues and educational pressure [13-15]. (Lines 55-57)

  • < References >

  1. Park, S.; Jang, H.; Lee, E.-S., Major Stressors among Korean Adolescents According to Gender, Educational Level, Residential Area, and Socioeconomic Status. International Journal of Environmental Research and Public Health 2018, 15 (10), 2080-2080.

  1. Lee, S.; Shouse, R. C., The Impact of Prestige Orientation on Shadow Education in South Korea. Sociology of Education 2011, 84 (3), 212-224.

  1. Statistics Korea Youth Statistics Available online: http://kostat.go.kr. (accessed on 27 September 2020).

  1. Lines 89-90 -- The authors seem to be contradicting themselves here as now they are talking about parental attitude toward psychiatric help for their children rather than for themselves. Which one did they explore in their survey?
  • Based on the previous studies, we hypothesized that parental attitude toward psychiatric help for their own mental health issue would influence the relationship among depression stigma, attitude toward psychiatric help, depression, and suicidal ideation of children. (Lines 93-96)

  1. Under Materials and Methods (lines 100-151), it can be more clearly stated which instruments were given to the parents and which to the children.
  • To parents, questionnaire of attitude toward psychiatric help and sample characteristic was given, on the other hand, questionnaire of suicidal ideation, depression, depression stigma, and attitude toward psychiatric help was distributed to their children. We added the description about distribution of questionnaires to parents and their children, and reorganized the order of description of instruments as you recommended.

For both parents and their children, the questionnaire of attitude toward psychiatric help was distributed. For adolescents, the questionnaire of suicidal ideation, depression, depression stigma was distributed. (Lines 117-119)

  1. I like the visual representation of the serial mediation analysis in Fig 1; it illustrates more clearly the ideas that the authors are trying to convey. Please note the typo ("ideatin") in the top right box.
  • Thank you for your comments. We revised the typo. (Page 7, Figure 1)

  1. Line 205 and line 241 -- "positively affects" and "positively influenced" are ambiguous as "positive" could have opposite meanings depending on how one reads the phrase. I would suggest reviewing the wording.
  • We revised the sentence to make it more clear. (Lines 211-212, 213-214)
  • As children’s stigma was high, their attitude toward psychiatric help was low (β = −0.256, p = 0.008). (Lines 211--212)
  • In addition, as depression of children was high, their suicidal ideation was also high(β = 0.804, p < 0.001). (Lines 213-214)

  1. Line 251 -- likewise, the meaning of "increasing adolescents' attitude" is unclear.
  • Making adolescents’ attitude toward psychiatric help more positively only (Line 263)

  1. Lines 252-254 -- The meaning of this sentence is unclear; I don't understand how reference 39 supports the conclusion that "parents should be educated about..."
  • Thank you for your comments. We added the better reference [48] supporting the conclusion.(Lines 264-267)
  • <Reference>
  1. Fazel, M., Hoagwood, K., Stephan, S., & Ford, T. (2014). Mental health interventions in schools 1: Mental health interventions in schools in high-income countries. The lancet. Psychiatry, 1(5), 377–387. https://doi.org/10.1016/S2215-0366(14)70312-8

  1. Lines 281-285 -- I think the authors have not adequately considered or discussed the limitations of their study here: they should include points such as how more than 90% of their study respondents (both the children and the parents) were female and also consider how sampling bias could influence their findings.
  • We reflected your suggestions in the limitation section. (Lines 292-298)

However, the generalizability of this study was limited since the data were collected in one middle school located in a southern part of South Korea, using convenience sampling. Second, most participants (both parents and their children) were female so that the findings of this study might be influenced by sampling bias. Therefore, for future study we need to recruit more fathers and sons. Fourth, all instruments were self-reported so that social desirability or recall bias may limit the study findings. Third, the study design was a cross-sectional, precluding explanation of causal relationships among variables. Fourth, this study measured the parental attitude toward psychiatric help for their own help, so that it might be limited to explore the parental attitude toward their children’s mental health issues.

  1. Lines 292-294 -- Can the authors cite references to support their statement about gender differences?
  • We added the reference [53] about gender differences on suicidal behaviors. (Lines 306-7)
  • <Reference>
  1. Miranda-Mendizabal, A., Castellví, P., Parés-Badell, O., Alayo, I., Almenara, J., Alonso, I., Blasco, M. J., Cebrià, A., Gabilondo, A., Gili, M., Lagares, C., Piqueras, J. A., Rodríguez-Jiménez, T., Rodríguez-Marín, J., Roca, M., Soto-Sanz, V., Vilagut, G., & Alonso, J. (2019). Gender differences in suicidal behavior in adolescents and young adults: systematic review and meta-analysis of longitudinal studies. International journal of public health, 64(2), 265–283. https://doi.org/10.1007/s00038-018-1196-1

  1. Lines 294-296 -- This content belongs in the discussion of the study limitations in the previous paragraph.
  • We moved this sentence to the study limitation as you recommended. (Line 295)

  1. Reference 39 seems to be unavailable in the databases I searched?
  • That article is in press so that it is not available yet.

Reviewer 3 Report

The very low numbers of boys in the study requires comment and discussion

Author Response

Responses to reviewer #3

Manuscript ID : ijerph-911067

Title: Influence of parental attitude toward psychiatric help on their children’s suicidal ideation: a serial mediation model

Comments and Suggestions for Authors

The very low numbers of boys in the study requires comment and discussion

  • Unfortunately, we were not able to collect many boys in this study, but we will add this variable for future research. Thank you for your valuable comments.
  • We reflected your suggestion in the discussion (Lines 292-297)

However, the generalizability of this study was limited since the data were collected in one middle school located in a southern part of South Korea, using convenience sampling. Second, most participants (both parents and their children) were female so that the findings of this study might be influenced by sampling bias. Therefore, for future study we need to recruit more fathers and sons.

Round 2

Reviewer 3 Report

none